# Multiview Data Clustering with Similarity Graph Learning Guided Unsupervised Feature Selection

**DOI:** 10.3390/e25121606

**Published:** 2023-11-30

**Authors:** Ni Li, Manman Peng, Qiang Wu

**Affiliations:** 1College of Information and Electronic Engineering, Hunan City University, Yiyang 413000, China; 2College of Information and Engineer, Hunan University, Changsha 410082, China; pengmanman@hnu.edu.cn (M.P.); wuqiang@hnu.edu.cn (Q.W.)

**Keywords:** multiview data clustering, unsupervised feature selection, similarity graph

## Abstract

In multiview data clustering, consistent or complementary information in the multiview data can achieve better clustering results. However, the high dimensions, lack of labeling, and redundancy of multiview data certainly affect the clustering effect, posing a challenge to multiview clustering. A clustering algorithm based on multiview feature selection clustering (MFSC), which combines similarity graph learning and unsupervised feature selection, is designed in this study. During the MFSC implementation, local manifold regularization is integrated into similarity graph learning, with the clustering label of similarity graph learning as the standard for unsupervised feature selection. MFSC can retain the characteristics of the clustering label on the premise of maintaining the manifold structure of multiview data. The algorithm is systematically evaluated using benchmark multiview and simulated data. The clustering experiment results prove that the MFSC algorithm is more effective than the traditional algorithm.

## 1. Introduction

Various application types correspond to various network attributes that describe individuals and groups from different perspectives. These networks are represented as multiview feature spaces. For example, when uploading photos to Flickr, users are required to offer labels and related text. In other words, photos can be represented by three view feature spaces: photo content, label, and text description spaces.

Multiview data can integrate these view spaces and use correlation to obtain more accurate network representations. Currently, multiview data are usually described in the form of graphs, such as Gaussian function graphs, k nearest neighbor graphs [1], and graphs based on subspace clustering [2,3]. For the selection of the correct neighborhood size and the processing of points near the intersection of the subspace, subspace clustering based on self-representation is superior to other graph-based representation methods. Nie et al. developed a multiview clustering [4,5] algorithm that can perform spectral clustering of an information network of multiple views by constructing a multiview similarity matrix. The multiview clustering algorithm [6] proposed by Bickel et al. uses spherical k-means multiview clustering. Pu et al. advanced the multiview clustering algorithm [7] based on matrix decomposition, which regularizes the similarity matrix using multiview manifold regularization, to merge the inherent and nonlinear structure of the network in every view. The aforementioned methods provide an idea regarding the relationships between multiview data that improve clustering performance [8] by constructing multiview similarity matrix clustering. However, the redundancy of multiview data has not yet been resolved. In addition, the calculation for constructing a multiview similarity matrix is complicated and unsuitable for large-scale multiview data.

Feature selection [9] obtains the low-dimensional feature subspace representation of the network by selecting features as well as removing noisy, irrelevant, and redundant features to preserve the inherent data structure. This is an effective method for handling large-scale high-dimensional networks. Most existing feature selection methods are based on single-view networks. Recently, the focus of unsupervised feature selection research has been on the study of multiview data. Zhang et al. [10] propose a formulation that learns an adaptive neighbor graph for unsupervised multiview feature selection. This formulation collaborates multiview features and discriminates between different views. Fang et al. [11] propose a novel approach that incorporates both cluster structure and a similarity graph. Their method utilizes multiview feature selection and an orthogonal decomposition technique, which breaks down each objective matrix into a base matrix and a clustering index matrix for each view. Cao et al. [12] present a cluster learning guided multiview unsupervised feature selection, which unified subspace learning, cluster learning, and feature selection into a framework. Tang et al. [13] propose a feature selection method based on multiview data that aims to maintain diversity and enhance consensus learning by utilizing cross-view local structures. Liu et al. [14] propose a framework for guided unsupervised feature selection, which utilizes consensus clustering to generate pseudo cluster indexes for the purpose of feature selection.

There are two modes of feature selection in multiview networks. One is the serial mode, which is a feature selection method that seriates the connection multiview feature space into a feature space and then selects the features. The other is the parallel mode, which involves performing traditional feature selection on each view simultaneously. In more detail, the serial mode ignores the differences between heterogeneous feature spaces, so its performance is relatively poor. The parallel mode considers the correlation between multiple view spaces with relatively better performance. Research on the unsupervised feature selection of multiview data without labels poses a significant challenge. For the traditional unsupervised feature selection method, the feature distribution selected by the Laplacian score [15] method agrees with the sample distribution, which can perform a good regional classification and reflect the inherent manifold structure of data. However, the correlation between the features is not evaluated, resulting in the selection of redundant features. In the MFSC method, spectral analysis retains the internal structure and $L_{2,1}$ uses feature selection coefficients to select the best features. Therefore, the selected features retain the clustering structure of the data.

The MFSC algorithm proposed in this study makes the following contributions:Compared with a single-view dataset that concatenates multiview data, the parallel use of multiview datasets from real-world social media sites significantly improves the accuracy of data representation.In integrated subspace clustering and feature selection, the clustering label and representative coefficient matrix are flow regularizations. Furthermore, to obtain a more suitable feature selection matrix, the a priori of the manifold structure is embedded in the feature selection model.In the construction of the parallel mode multiview feature selection algorithm, noisy, irrelevant, and redundant features are removed to preserve the inherent data structure and improve the efficiency and quality of feature selection based on clustering, which is more suitable for multiview data.

The rest of this paper is organized as follows. Section 2 introduces the basic studies related to the MFSC algorithm. Section 3 presents the MFSC model and its optimization iterative process in detail, and it theoretically proves the convergence and complexity of the algorithm. Section 4 reports the parameter sensitivity and performance analysis of MFSC on typical datasets, as well as the results of comparison experiments with some single-view or multiview feature selections. Section 5 presents the results of this study and the future work.

## 2. Related Studies

### 2.1. Multiview Subspace Representation

Let $X^{\left(v\right)}$ be the data sample node of the *v*-th view and $S^{\left(v\right)}$ be its representative coefficient matrix. Each data point in the subspace union can be reconstructed effectively by combining the other points in the dataset. Given the data X based on the group effect [16], for representation coefficients $S_{i}$ and $S_{j}$ of the samples, $X_{i}$ and $X_{j}$ are similar and so are $S_{i}$ and $S_{j}$. The multiview representation of traditional sparse subspace clustering (MVSC) [17] is defined as follows:(1)$$\left|\right|X^{\left(v\right)}-X^{\left(v\right)}S^{\left(v\right)}{\left|\right|}_{F}^{2},s.t.S^{\left(v\right)}I=I,S^{\left(v\right)}\left(i,i\right)=0.$$

MVSC can well capture the self-representation matrix in the multisubspace k-nearest neighbor graph structure. Similar structure graph $Z^{\left(v\right)}=\frac{{\left(S^{\left(v\right)}\right)}^{T}+S^{\left(v\right)}}{2}$ of the *v*-th view can learn the multisubspace structure when there are noise, abnormal values, and damaged or missing entries in the data.

### 2.2. Multiview Unsupervised Feature Selection

Most existing studies on multiview learning [18] assume that all views share the same label space and that these views are related to each other through the label space. It is well known that the main difficulty of unsupervised feature selection is the lack of class tags. Consequently, the concept of a pseudo-class label is introduced to guide the development of the framework using the relationship between views, which is defined as follows:(2)$$\left|\right|{\left(X^{\left(v\right)}\right)}^{T}W^{\left(v\right)}-{C||}_{F}^{2}+\alpha ||W^{\left(v\right)}{\left|\right|}_{2,1},$$
where the *v*-th view has a mapping matrix $W^{\left(v\right)}$ that assigns the pseudo-class label *C* to the data points. Based on the assumption that the view is associated with the shared label space, each pseudo-class label allocation matrix ${\left(X^{\left(v\right)}\right)}^{T}W^{\left(v\right)}$ is approximated such that it is close to the pseudo-class label matrix. The $l_{2,1}$ norm [19] is added to $W_{i}$ to ensure sparseness in the $W_{i}$ row and feature selection. In addition, the $l_{2,1}$ norm is convex, making the optimization easier.

### 2.3. Multiview Manifold Structure

The greater the similarity value of the two data points, the more similar the clusters. A similar structure graph with k unconnected cluster subspaces can be directly learned and it is defined as follows:(3)$$\sum_{i,j=1}^{n}\left|\right|C_{i,:}-C_{j,:}{\left|\right|}_{F}^{2}S_{ij}^{\left(v\right)}=tr\left(C^{T}L^{\left(v\right)}C\right),$$
where clustering label $C\in R^{n\times k}$ and Laplacian matrix $L^{\left(v\right)}=D^{\left(v\right)}-\frac{{\left(S^{\left(v\right)}\right)}^{T}+S^{\left(v\right)}}{2}$. It is known that MHOAR [20] points out that the properties of the *L* matrix of nonnegative matrix *S* are shown in Theorem 1.

**Theorem** **1.**
*The number of the eigenvalues 0 of normalized L is equal to the number of connected subspaces of S. Therefore, rank(L)=n−k. According to the Ky Fan theorem [21], using $\sigma_{i}\left(L\right)$ to represent the i-th smallest eigenvalue of L, then $\sigma_{i}\left(L\right)\geq 0$ and rank(L)=n−k. Therefore, $\sum_{i=1}^{k}\sigma_{i}\left(L\right)=\arg min_{C\in R^{n\times k},C^{T}C=I_{k}}tr\left(C^{T}LC\right)$.*


## 3. Proposed Model

This section contains an introduction to the MFSC model: an explanation of the iterative optimization implementation process, algorithm, proof of convergence, and analysis of algorithm complexity. An illustration of the MFSC model is shown in Figure 1. Multiview unsupervised feature selection, similarity graph learning, and clustering index learning are achieved in the parallel mode. MFSC reduces the redundancy and irrelevant influence of multiview data and uses the clustering index as the feature selection standard to ensure that the clustering structure remains unchanged.

### 3.1. MFSC Model

Suppose the dataset $X={\left\{X^{\left(v\right)}\in R^{d^{\left(v\right)}\times n}\right\}}_{v=1}^{m}$ denotes the data of the *v*-th view, $d^{\left(v\right)}$ denotes the feature number of the *v*-th view, and *n* denotes the number of data. The feature selection matrix is $W^{\left(v\right)}\in R^{d^{\left(v\right)}\times k}$, the clustering label is $C\in R^{n\times k}$, and the subspace representation coefficient $S^{\left(v\right)}\in R^{n\times n}$, where *k* denotes the cluster number. The MFSC model is defined as follows:(4)$$\begin{array}{cc} \arg min_{S^{\left(v\right)},C,W^{\left(v\right)}} & \sum_{v=1}^{m}u^{\left(v\right)}\left(|\right|{\left(X^{\left(v\right)}\right)}^{T}W^{\left(v\right)}-C{\left|\right|}_{F}^{2}\\ \; & +\left|\right|X^{\left(v\right)}-X^{\left(v\right)}S^{\left(v\right)}{\left|\right|}_{F}^{2}+\alpha tr\left(C^{T}L^{\left(v\right)}C\right)\\ \; & +\beta ||W^{\left(v\right)}{\left|\right|}_{2,1})\\ & s.t.C^{T}C=I_{k},S^{\left(v\right)}I=I,S^{\left(v\right)}\left(i,i\right)=0, \end{array}$$
where $L^{\left(v\right)}=D^{\left(v\right)}-\frac{{\left(S^{\left(v\right)}\right)}^{T}+S^{\left(v\right)}}{2}$.

The model independently learns the $S^{\left(v\right)}$ of each view instead of directly using the $S^{\left(v\right)}$ calculated by the kernel function. Using the similarity graph of self-representation learning based on the manifold structure, the multisubspace structure of the data can be effectively reflected. By integrating subspace similarity graph learning and feature selection, the pseudo-class label C can capture the relationship between the views to obtain a robust and clean pseudo-class label. Row sparsity is achieved by applying the $l_{2,1}$ [22] constraint to $W^{\left(v\right)}$. Figure 1 shows the feature selection based on the parallel mode that iteratively updates the similarity matrix ${\left\{S^{\left(v\right)}\right\}}_{v=1}^{m}$, the feature selection matrix ${\left\{W^{\left(v\right)}\right\}}_{v=1}^{m}$, and the pseudo label matrix *C*.

### 3.2. Optimization Calculation Process and Algorithm Representation

This section first introduces the effective implementation of the iterative method to solve the optimization calculation in Equation (Equation 4). In the implementation process, $W^{\left(v\right)}$, $S^{\left(v\right)}$, and *C* are updated iteratively to obtain the specific implementation process of the MFSC algorithm.


**Update $W^{\left(v\right)}$:**


To effectively calculate the feature selection matrix $W^{\left(v\right)}$, irrelevant items $S^{\left(v\right)}$ and *C* are fixed. The objection equation can be rewritten as follows:(5)$$J1\left(W^{\left(v\right)}\right)=\left|\right|{\left(X^{\left(v\right)}\right)}^{T}W^{\left(v\right)}-{C||}_{F}^{2}+\beta ||W^{\left(v\right)}{\left|\right|}_{2,1}.$$

Given that this equation is nondifferentiable [19], the equation is transformed into:(6)$$J1\left(W^{\left(v\right)}\right)=\left|\right|{\left(X^{\left(v\right)}\right)}^{T}W^{\left(v\right)}-C{\left|\right|}_{F}^{2}+\beta tr\left({\left(W^{\left(v\right)}\right)}^{T}D^{\left(v\right)}W^{\left(v\right)}\right),$$
where $D^{\left(v\right)}$ denotes a diagonal matrix and the *j*-th diagonal element is $D^{\left(v\right)}\left(j,j\right)=\frac{1}{2||W_{j}^{\left(v\right)}{\left|\right|}_{2}}$.

Calculation process:(7)$$\begin{array}{cc} \frac{\partial J1(W^{\left(v\right)})}{\partial W^{\left(v\right)}} & =2X^{\left(v\right)}{\left(X^{\left(v\right)}\right)}^{T}W^{\left(v\right)}-2X^{\left(v\right)}C\\ \; & +2\beta D^{\left(v\right)}W^{\left(v\right)}=0, \end{array}$$

The updated rules for $W^{\left(v\right)}$ are as follows:(8)$$W^{\left(v\right)}={\left(X^{\left(v\right)}{\left(X^{\left(v\right)}\right)}^{T}+\beta D^{\left(v\right)}\right)}^{-1}X^{\left(v\right)}C.$$


**Update $S^{\left(v\right)}$:**


**Theorem** **2.**
*Given $X=XZ,W=\frac{Z+Z^{T}}{2}$, and L=D−W, then*

(9)
$$tr\left(F^{T}LF\right)=\frac{1}{2}tr\left(WP\right),$$

*with $P_{ij}=\left|\right|f_{i}-f_{j}{\left|\right|}_{2}^{2}$, where $f_{i}$ is the i-th row vector of matrix F.*


To effectively calculate the clustering label *C*, irrelevant items $W^{\left(v\right)}$ and $S^{\left(V\right)}$ are fixed. The objection equation can be rewritten as follows:(10)$$\begin{array}{cc} J2(S^{\left(v\right)}) & =\left|\right|X^{\left(v\right)}-X^{\left(v\right)}S^{\left(v\right)}{\left|\right|}_{F}^{2}+\alpha tr\left(C^{T}L^{\left(v\right)}C\right)\\ \; & +\rho ||I^{T}-I^{T}S^{\left(v\right)}{\left|\right|}_{F}^{2}\\ \; & s.t.\;S^{\left(v\right)}\left(i,i\right)=0. \end{array}$$

Based on the properties of the matrix trace $tr\left(X^{T}Y\right)=tr\left(XY^{T}\right)$ and Theorem 2, it is known that *P* is a symmetric matrix; then,
(11)$$\begin{array}{cc} tr(C^{T}L^{\left(v\right)}C) & =\frac{1}{2}tr\left(WP\right)\\ \; & =\frac{1}{2}\left(\frac{1}{2}tr\left(S^{\left(v\right)}P^{T}\right)+\frac{1}{2}tr\left({\left(S^{\left(v\right)}\right)}^{T}P\right)\right)\\ \; & =\frac{1}{2}tr\left({\left(S^{\left(v\right)}\right)}^{T}P\right), \end{array}$$
where $P_{ij}=\left|\right|C_{i}-C_{j}{\left|\right|}_{2}^{2}$. According to
(12)$$\begin{array}{cc} \; & \left|\right|X^{\left(v\right)}-X^{\left(v\right)}S^{\left(v\right)}{\left|\right|}_{F}^{2}+\rho ||I^{T}-I^{T}S^{\left(v\right)}{\left|\right|}_{F}^{2}\\ \; & =\left|\right|{\left[{\left(X^{\left(v\right)}\right)}^{T},p*I\right]}^{T}-{\left[{\left(X^{\left(v\right)}\right)}^{T},p*I\right]}^{T}S^{\left(v\right)}{\left|\right|}_{F}^{2}, \end{array}$$
suppose $X_{1}^{\left(v\right)}={\left[{\left(X^{\left(v\right)}\right)}^{T},p*I\right]}^{T}$, so $J2(S^{\left(v\right)})$ is equivalently expressed as follows:(13)$$\begin{array}{cc} J2(S^{\left(v\right)}) & =\left|\right|X_{1}^{\left(v\right)}-X_{1}^{\left(v\right)}S^{\left(v\right)}{\left|\right|}_{F}^{2}+\frac{\alpha }{2}tr\left({\left(S^{\left(v\right)}\right)}^{T}P\right)\\ \; & s.t.\;S^{\left(v\right)}\left(i,i\right)=0. \end{array}$$

Given that $tr\left({\left(S^{\left(v\right)}\right)}^{T}P\right)=\sum_{i}S^{\left(v\right)}\left(i,:\right)P\left(:,i\right)$, where $S^{\left(v\right)}\left(i,:\right)$ denotes the *i*-th row vector of $S^{\left(v\right)}$ and P(:,i) denotes the *i*-th column vector of *P*. Then,
(14)$$\begin{array}{cc} \; & \left|\right|X_{1}^{\left(v\right)}-X_{1}^{\left(v\right)}S^{\left(v\right)}{\left|\right|}_{F}^{2}=\\ \; & tr{\left(X_{1}^{\left(v\right)}-X_{1}^{\left(v\right)}S^{\left(v\right)}\right)}^{T}\left(X_{1}^{\left(v\right)}-X_{1}^{\left(v\right)}S^{\left(v\right)}\right)\\ \; & =\sum_{i}\left(X_{1}^{\left(v\right)}\left(i,:\right)-X_{1}^{\left(v\right)}S^{\left(v\right)}\left(i,:\right)\right)\\ \; & {\left(X_{1}^{\left(v\right)}\left(i,:\right)-X_{1}^{\left(v\right)}S^{\left(v\right)}\left(i,:\right)\right)}^{T}. \end{array}$$

Suppose $X_{s}^{\left(v\right)}=X_{1}^{\left(v\right)}-\left(X_{1}^{\left(v\right)}S^{\left(v\right)}-X_{1}^{\left(v\right)}\left(:,i\right)S^{\left(v\right)}\left(i,:\right)\right)$; then,
(15)$$\begin{array}{cc} \; & \left|\right|X_{1}^{\left(v\right)}-X_{1}^{\left(v\right)}S^{\left(v\right)}{\left|\right|}_{F}^{2}=\left|\right|X_{s}^{\left(v\right)}-X_{1}^{\left(v\right)}\left(:,i\right)S^{\left(v\right)}\left(i,:\right){\left|\right|}_{F}^{2}\\ \; & =tr({\left(X_{s}^{\left(v\right)}-X_{1}^{\left(v\right)}\left(:,i\right)S^{\left(v\right)}\left(i,:\right)\right)}^{T}\left(X_{s}^{\left(v\right)}-X_{1}^{\left(v\right)}\left(:,i\right)S^{\left(v\right)}\left(i,:\right)\right))\\ \; & =tr({\left(X_{s}^{\left(v\right)}\right)}^{T}X_{s}^{\left(v\right)}-{\left(S^{\left(v\right)}\left(i,:\right)\right)}^{T}{\left(X_{1}^{\left(v\right)}\left(:,i\right)\right)}^{T}X_{s}^{\left(v\right)}\\ \; & -{\left(X_{s}^{\left(v\right)}\right)}^{T}X_{1}^{\left(v\right)}\left(:,i\right)S^{\left(v\right)}\left(i,:\right)\\ \; & +{\left(S^{\left(v\right)}\left(i,:\right)\right)}^{T}{\left(X_{1}^{\left(v\right)}\left(:,i\right)\right)}^{T}X_{1}^{\left(v\right)}\left(:,i\right)S^{\left(v\right)}\left(i,:\right))\\ \; & =tr\left({\left(X_{s}^{\left(v\right)}\right)}^{T}X_{s}^{\left(v\right)}\right)-2tr\left(S^{\left(v\right)}\left(i,:\right){\left(X_{s}^{\left(v\right)}\right)}^{T}X_{1}^{\left(v\right)}\left(:,i\right)\right)\\ \; & +tr({\left(X_{1}^{\left(v\right)}\left(:,i\right)\right)}^{T}X_{1}^{\left(v\right)}\left(:,i\right)S^{\left(v\right)}\left(i,:\right){\left(S^{\left(v\right)}\left(i,:\right)\right)}^{T}). \end{array}$$

Subsequently, the objective vector expression for $S^{\left(v\right)}\left(i,:\right)$ is obtained as follows:(16)$$\begin{array}{cc} J2(S^{\left(v\right)}\left(i,:\right))= & {\left(X_{1}^{\left(v\right)}\left(:,i\right)\right)}^{T}X_{1}^{\left(v\right)}\left(:,i\right)S^{\left(v\right)}\left(i,:\right){\left(S^{\left(v\right)}\left(i,:\right)\right)}^{T}\\ \; & -2S^{\left(v\right)}\left(i,:\right){\left(X_{s}^{\left(v\right)}\right)}^{T}X_{1}^{\left(v\right)}\left(:,i\right)\\ \; & +\frac{\alpha }{2}S^{\left(v\right)}\left(i,:\right)P\left(:,i\right). \end{array}$$

Similarly, the objective vector expression can also be expressed in the following form. There is only a constant difference between the two forms:(17)$$\begin{array}{cc} J2(S^{\left(v\right)}\left(i,:\right))= & \left|\right|{\left(S^{\left(v\right)}\left(i,:\right)\right)}^{T}-\frac{{\left(X_{s}^{\left(v\right)}\right)}^{T}X_{1}^{\left(v\right)}\left(:,i\right)}{{\left(X_{1}^{\left(v\right)}\left(:,i\right)\right)}^{T}X_{1}^{\left(v\right)}\left(:,i\right)}{\left|\right|}_{2}^{2}\\ \; & +\frac{\alpha }{2}S^{\left(v\right)}\left(i,:\right)P\left(:,i\right)\\ \; & s.t.\;S^{\left(v\right)}\left(i,:\right)\left(i\right)=0. \end{array}$$

Let $\frac{{\left(X_{s}^{\left(v\right)}\right)}^{T}X_{1}^{\left(v\right)}\left(:,i\right)}{{\left(X_{1}^{\left(v\right)}\left(:,i\right)\right)}^{T}X_{1}^{\left(v\right)}\left(:,i\right)}=Q^{\left(v\right)}\left(:,i\right)$. Suppose the subscript of the vector is *k*; then, if k=i, ${\left(S^{\left(v\right)}\left(i,:\right)\right)}^{T}\left(k\right)=0$, otherwise solve the following equation:(18)$$\begin{array}{cc} J2(S^{\left(v\right)}\left(i,:\right)\left(k\right))= & {\left({\left(S^{\left(v\right)}\left(i,:\right)\right)}^{T}\left(k\right)-Q^{\left(v\right)}\left(:,i\right)\left(k\right)\right)}^{2}\\ \; & +\frac{\alpha }{2}S^{\left(v\right)}\left(i,:\right)\left(k\right)P\left(:,i\right)\left(k\right), \end{array}$$

According to $\frac{dJ2(S^{\left(v\right)}\left(i,:\right)\left(k\right))}{dS^{\left(v\right)}\left(i,:\right)\left(k\right)}=0$, the solution is as follows:  
(19)$${\left(S^{\left(v\right)}\left(i,:\right)\right)}^{T}\left(k\right)=\left\{\begin{array}{cc} \; & Q^{\left(v\right)}\left(:,i\right)\left(k\right)-\frac{\alpha P(:,i)(k)}{4},\\ \; & if\;Q^{\left(v\right)}\left(:,i\right)\left(k\right)>\frac{\alpha P(:,i)(k)}{4};\\ \; & Q^{\left(v\right)}\left(:,i\right)\left(k\right)+\frac{\alpha P(:,i)(k)}{4},\\ \; & if\;Q^{\left(v\right)}\left(:,i\right)\left(k\right)<\frac{\alpha P(:,i)(k)}{4};\\ \; & 0,otherwise. \end{array}\right.$$


**Update *C*:**


To effectively calculate the clustering label *C*, $W^{\left(v\right)}$ and $S^{\left(V\right)}$ are fixed, and irrelevant items are ignored. The optimization formula can be rewritten as follows:(20)$$\begin{array}{cc} J3(C)= & \sum_{v=1}^{m}u^{\left(v\right)}\left(|\right|{\left(X^{\left(v\right)}\right)}^{T}W^{\left(v\right)}-{C||}_{F}^{2}+\alpha tr\left(C^{T}L^{\left(v\right)}C\right)),\\ \; & s.t.C^{T}C=I_{k},C\geq 0. \end{array}$$

To remove the orthogonal constraint, a penalty term $p||C^{T}C-I_{k}{\left|\right|}_{F}^{2}$ is added to function (Equation 20). The following optimization functions are available:(21)$$\begin{array}{cc} \; & \sum_{v=1}^{m}u^{\left(v\right)}\left(|\right|{\left(X^{\left(v\right)}\right)}^{T}W^{\left(v\right)}-{C||}_{F}^{2}+\alpha tr\left(C^{T}L^{\left(v\right)}C\right))\\ \; & +\rho ||C^{T}C-I_{k}{\left|\right|}_{F}^{2},s.t.C\geq 0. \end{array}$$

The Lagrangian operator ϕ is introduced to remove the inequality constraints and the following Lagrangian function is obtained:(22)$$\begin{array}{cc} \omega (C,\phi )= & \sum_{v=1}^{m}u^{\left(v\right)}\left(|\right|{\left(X^{\left(v\right)}\right)}^{T}W^{\left(v\right)}-{C||}_{F}^{2}+\alpha tr\left(C^{T}L^{\left(v\right)}C\right))\\ \; & +\rho ||C^{T}C-I_{k}{\left|\right|}_{F}^{2}-tr\left(\phi^{T}C\right). \end{array}$$

Take ω(C,ϕ) to the derivative of *C*, then:(23)$$\begin{array}{cc} \frac{\partial \omega (C,\phi )}{\partial C}= & \sum_{v=1}^{m}u^{\left(v\right)}\left(-2{\left(X^{\left(v\right)}\right)}^{T}W^{\left(v\right)}+2C+2\alpha L^{\left(v\right)}C\right)\\ \; & +4\rho C(C^{T}C-I_{k})-\phi =0. \end{array}$$

Thus, ϕ is obtained as follows:(24)$$\begin{array}{cc} \phi = & \sum_{v=1}^{m}u^{\left(v\right)}\left(-2{\left(X^{\left(v\right)}\right)}^{T}W^{\left(v\right)}+2C+2\alpha L^{\left(v\right)}C\right)\\ \; & +4\rho C(C^{T}C-I_{k}). \end{array}$$

Based on the Karush–Kuhn–Tucker condition [23] $\phi_{ij}C_{ij}=0$, the following equation is obtained:(25)$$\begin{array}{c} (\sum_{v=1}^{m}u^{\left(v\right)}\left(-2{\left(X^{\left(v\right)}\right)}^{T}W^{\left(v\right)}+2C+2\alpha L^{\left(v\right)}C\right)\\ +4\rho C\left(C^{T}C-I_{k}\right){)}_{ij}C_{ij}=0. \end{array}$$

The following update formulas are obtained:(26)$$\begin{array}{c} C_{ij}=\frac{{\left(\sum_{v=1}^{m}u^{\left(v\right)}\left({\left(X^{\left(v\right)}\right)}^{T}W^{\left(v\right)}\right)+2\rho C\right)}_{ij}}{{\left(\sum_{v=1}^{m}u^{\left(v\right)}\left(C+\alpha L^{\left(v\right)}C\right)+2\rho CC^{T}C\right)}_{ij}}C_{ij}. \end{array}$$

After updating *C*, *C* must be regularized to ensure that it satisfies the following constraint: $C^{T}C=I_{k}$.

### 3.3. Convergence

**Theorem** **3.**
*The iterative optimization process $J1(W^{\left(v\right)})$ automatically reduces the objective function value until it converges.*


**Proof.** The first term of $J1\left(W^{\left(v\right)}\right):\Omega \left(X^{\left(v\right)},W^{\left(v\right)},C\right)=\left|\right|{\left(X^{\left(v\right)}\right)}^{T}W^{\left(v\right)}-C{\left|\right|}_{F}^{2}$ and its Hessian matrix value is:
(27)$$\frac{\partial^{2}\Omega \left(X^{\left(v\right)},W^{\left(v\right)},C\right)}{\partial {\left(W^{\left(v\right)}\right)}^{2}}=2X^{\left(v\right)}{\left(X^{\left(v\right)}\right)}^{T}\geq 0.$$Therefore, $J1(W^{\left(v\right)})$ is convex; that is,
(28)$$\Omega \left(X^{\left(v\right)},W_{\left(t+1\right)}^{\left(v\right)},C\right)\leq \Omega \left(X^{\left(v\right)},W_{\left(t\right)}^{\left(v\right)},C\right).$$The second term of $J1(W^{\left(v\right)})$:$\Phi \left(W^{\left(v\right)}\right)=\left|\right|W^{\left(v\right)}{\left|\right|}_{2,1}=\sum_{i=1}^{l^{\left(v\right)}}\sqrt{\sum_{j=1}^{k}{\left(W_{ij}^{\left(v\right)}\right)}^{2}}=\sum_{i=1}^{l^{\left(v\right)}}\left|\right|W_{i}^{\left(v\right)}{\left|\right|}_{2}$ and its Hessian matrix value is
(29)$$\frac{\partial^{2}tr\left({\left(W^{\left(v\right)}\right)}^{T}D^{\left(v\right)}W^{\left(v\right)}\right)}{\partial {\left(W^{\left(v\right)}\right)}^{2}}=2D^{\left(v\right)}\geq 0.$$Therefore,
(30)$$\Phi \left(W_{t+1}^{\left(v\right)}\right)\leq \Phi \left(W_{t}^{\left(v\right)}\right).$$Then, $J1\left(W_{\left(t+1\right)}^{\left(v\right)}\right)\leq J1\left(W_{\left(t\right)}^{\left(v\right)}\right)$. The proof is completed.    □

**Theorem** **4.**
*The iterative optimization process of Algorithm 1 automatically reduces the value of the objective function (Equation 4) until it converges.*


**Proof.** Other variables are fixed such that the objective function $J1(W^{\left(v\right)})$ is related to $W^{\left(v\right)}$. Theorem 3 proves that, under the update rule, the objective value of $J1(W^{\left(v\right)})$ is automatically reduced:
(31)$$J\left(S^{\left(v\right)},C,W_{t+1}^{\left(v\right)}\right)\leq J\left(S^{\left(v\right)},C,W_{t}^{\left(v\right)}\right).$$Given the other fixed variables, the objective function $J2(W^{\left(v\right)})$ is related to $S^{\left(v\right)}$. Then, the Hessian matrix of $J2(W^{\left(v\right)})$ is $\frac{\partial^{2}J2\left(S^{\left(v\right)}\right)}{\partial {\left(S^{\left(v\right)}\right)}^{2}}=2{\left(X_{1}^{\left(v\right)}\right)}^{T}X_{1}^{\left(v\right)}\geq 0$, which is a positive semi-definite matrix. Therefore,
(32)$$J\left(S_{t+1}^{\left(v\right)},C,W^{\left(v\right)}\right)\leq J\left(S_{t}^{\left(v\right)},C,W^{\left(v\right)}\right).$$Fix other variables and update *C*; the Hessian matrix of the objective function J3(C) is $2\tilde{L^{\left(v\right)}}\geq 0$, where $\tilde{L^{\left(v\right)}}=aL^{\left(v\right)}+I$. Thus,
(33)$$J\left(S^{\left(v\right)},C_{t+1},W^{\left(v\right)}\right)\leq J\left(S^{\left(v\right)},C_{t},W^{\left(v\right)}\right).$$The proof is complete.    □

**Algorithm 1** MFSC FOR CLUSTERING**Require:** {X(v)}v=1m,{u(v)}v=1m,k,α,β**Ensure:** ACC,NMI **for** v=1 to m **do**  initialize {S(v)}v=1m, {W(v)}v=1m, and *C* **end for** **while** not convergence **do**  **for** v=1 to m **do**   update {W(v)}v=1m according to Equation (Equation 8)   update {S(v)}v=1m according to Equation (Equation 19)   update *C* according to Equation (Equation 26)  **end for** **end while** **for** *v* = 1 to *m* **do**  Sort each feature for X(v) according to ||W(v)||F2 in descending order and select the top-*f* ranked ones;  X_new=[X_new,Xf(v)] **end for** kmeans clustering for X_new; Calculate ACC and NMI


### 3.4. Complexity Analysis

In this section, the time complexity of the three subproblems in the optimization model is calculated:

In subproblem $J1(W^{\left(v\right)})$, term $X^{\left(v\right)}{\left(X^{\left(v\right)}\right)}^{T}$ requires $O(n{\left(d^{\left(v\right)}\right)}^{2})$ and its inverse matrix requires $O(n{\left(d^{\left(v\right)}\right)}^{3})$. The time complexity of term ${\left(X^{\left(v\right)}{\left(X^{\left(v\right)}\right)}^{T}+\beta D^{\left(v\right)}\right)}^{-1}X^{\left(v\right)}C$ is $O(d^{\left(v\right)}\times n\times k)$. Therefore, the total time complexity of the subproblem is $O(\sum_{v=1}^{m}\left(n{\left(d^{\left(v\right)}\right)}^{2}+n{\left(d^{\left(v\right)}\right)}^{3}+d^{\left(v\right)}\times n\times k\right))$.

In subproblem $J2(S^{\left(v\right)})$, each row of $S^{\left(v\right)}$ requires matrix multiplication and the time complexity is $O(n^{2}d^{\left(v\right)})$. Therefore, the total time complexity of the subproblem is $O(\sum_{v=1}^{m}n^{3}d^{\left(v\right)})$.

In subproblem J3(C), the calculation of term ${\left(X^{\left(v\right)}\right)}^{T}W^{\left(v\right)}$ requires $O(d^{\left(v\right)}\times n\times k)$, and the calculation of terms $L^{\left(v\right)}C$ and $CC^{T}C$ requires $O(n^{2}k)$. The total time complexity is $O(\sum_{v=1}^{m}\left(d^{\left(v\right)}\times n\times k+n^{2}\times k\right))$.

## 4. Experiment

This section conducts an evaluation experiment on the MFSC algorithm using some kinds of benchmark multiview datasets and its performance is compared with those of other related algorithms.

### 4.1. Dataset

The evaluation experiment of the MFSC algorithm was conducted on 5 real multiview datasets: news dataset 3sources, paper dataset Cora, information retrieval and research dataset CiteSeer, website dataset BBCSport, and blog website dataset BlogCatalog. Table 1 summarizes the 5 datasets. In addition, the specific information is as follows:**3sources** The news dataset comes from three online news sources: BBC, Reuters, and Guardian. All articles are placed within the text. Out of a total of 948 articles from three sources, 169 are adopted. It is noteworthy that each article in the dataset has a main theme.**Cora** The paper dataset contains a total of 2708 sample points, which are divided into 7 categories. Each sample point is a scientific paper. A paper comprises a 1433-dimensional word vector.**CiteSeer** The papers in the information retrieval and research dataset are divided into six categories, containing a total of 3312 papers, and records the citation or citation information between the papers. Through sorting, 3703 unique words are obtained.**BBCSport** The website dataset comes from 544 dataset points of the BBC sports website, including sports news related to 5 subject areas (athletics, cricket, football, rugby, and tennis) and 2 related views.**BlogCatalog** BlogCatalog is the social blog directory which manages the bloggers and their blogs. The data consists of 10,312 articles, divided into 6 categories, each article with two views: blog content and its related tags.

### 4.2. Benchmark Method

MFSC is compared with the following algorithms.

LapScore (the Laplacian score function) selects features with strong separability, where the distribution of feature vector values is consistent with the sample distribution, thereby reflecting the inherent manifold structure of the data.Relief is a multiclass feature selection algorithm. The larger the weight of the feature, the stronger the classification ability of the feature. Features with weights less than a certain threshold are removed.MCFS [24] (a multiclustering feature selection) algorithm uses the spectral method to preserve the local manifold topology and selects features using a method that can preserve the clustering topology.PRMA [7] (probabilistic robust matrix approximation) is a multiview clustering algorithm with robust regularization matrix approximation. Powerful norm and manifold regularization are used for regularization matrix factorization, making the model more distinguishable in multiview data clustering.GMNMF [17] (graph-based multiview nonnegative matrix factorization) is a multiview nonnegative matrix decomposition clustering algorithm involving intrinsic structure information among multiview graphs.SCFS [3] (subspace clustering-based feature selection) is an unsupervised feature selection method based on subspace clustering that maintains a similarity relation by learning the representation of the low-dimensional subspace of samples.JMVFG [11] (joint multiview unsupervised feature selction and graph leaning) proposed a unified objective function that can simultaneously learn clustering structure, and global and local similarity graphs.CCSFS [12] (consensus cluster structure guided multiview unsupervised feature selection) unifies subspace learning, clustering learning, consensus learning, and unsupervised feature selection into one optimization framework for mutual optimization.

### 4.3. Evaluation Metrics

**ACC (Accuracy)** is used to compare the obtained cluster labels $cluster\_label_{i}$ with the real cluster labels $truth\_label_{i}$. The ACC is defined as follows:(34)$$ACC=\frac{\sum (cluster\_label_{i}==truth\_label_{i})}{m},$$
where *m* denotes the total number of data samples.

**NMI (Normalized Mutual Information)** is the mutual information entropy between the obtained and real cluster labels; it is defined as follows:(35)$$NMI=\frac{\sum_{i=1}^{K}\sum_{i=1}^{K}n_{i,j}log\left(\frac{n{\dot{n}}_{i,j}}{n_{i}n_{j}}\right)}{\sqrt{\left(\sum_{i=1}^{K}n_{i}log\left(\frac{n_{i}}{n}\right)\right)\left(\sum_{j=1}^{K}n_{j}log\left(\frac{n_{j}}{n}\right)\right)}},$$
where $n_{i}$ denotes the sample number of cluster $C_{i}\;\left(1\leq i\leq K\right)$ and $n_{i,j}$ denotes the sample number in both cluster $C_{i}$ and category $C_{j}$.

### 4.4. Parameter Setting

The MFSC algorithm has two main parameters α and β. In the experiment, the parameter range of α is set to $\left\{10^{-3},10^{-2},0.1,1,10,10^{2},10^{4}\right\}$ and that of β is set to $\left\{10^{-6},10^{-3},10^{-2},1,10,10^{2},10^{4}\right\}$. The correlation coefficient of the 3sources data is u={0.3,0.3,0.4} and the other data views are the two-view data, with the correlation coefficient defined as follows u={0.5,0.5}. The value range of feature# (feature selection number) is set to {100,200,300,400,500}. Due to the large scale of BlogCatalog dataset, its range is feature#∈{500,1000,1500,2000,2500}. Considering that the clustering method k-means usually converges to a local minimum, it is necessary to repeat each experiment 20 times and report the average performance.

### 4.5. Results of Multiview Clustering

Table 2 and Table 3 show the ACC and NMI values of the different feature selection and multiview clustering methods. To determine the impact of the benchmark feature selection method on clustering, this experiment first merges the results of multiview feature selection into new data and then executes k-means. The final value is the average value of the clustering of different feature selection values. Based on the experimental results, MFSC performs well on both ACC and NMI, which proves the effectiveness of the algorithm.

### 4.6. Parameter Analysis

To achieve peak clustering performance, we tune parameters α, β, and feature#. Thus, we alter their values to see how they affect the ACC and NMI of clustering for 3sources data, Cora data, CiteSeer data, BBCSport data and BlogCatalog data.

Figure 2, Figure 3 and Figure 4 show the clustering experiment results of parameters α, β, and feature# in the 3sources dataset.

Figure 2 shows the change description of α, β, and the clustering indexes ACC and NMI in 3source. The average value is taken as the final result. Based on the ACC and NMI results in 3source, the MFSC algorithm is sensitive to parameters α and β. When parameter α is small, the performance of ACC is relatively high. When parameter β is large, the performance of NMI is relatively high.

Figure 3 shows the change description of parameter α and feature# and the values of clustering indexes ACC and NMI from 3source. In most cases, when the parameter α=0.001, the ACC and NMI of the MFSC exhibit better performance in the feature selection dimension, which shows the importance of capturing the multiview manifold structure and embedding it into the feature selection model.

Figure 4 shows the change description of parameter β and feature number and the clustering performance ACC and NMI values from 3source. It can be concluded that the MFSC is sensitive to the selected feature number. As the value of feature selection increases, the ACC and NMI increase. In most cases, when the parameter β = 10,000, the ACC and NMI of the MFSC exhibit better performance. To ensure the sparsity of matrix W, the larger the value of the feature selection, the greater the importance and the stronger the clustering performance.

Figure 5, Figure 6 and Figure 7 show the clustering results of parameters α, β, and feature# in the Cora dataset.

Figure 5a shows that the ACC is insensitive to parameter α and insensitive to parameter β on interval β∈[0.000001,0.001] or β∈[0.01,1,10,100,10000]. Figure 5b shows that the NMI is insensitive to parameter α but is sensitive to parameter β. When β≥1, the NMI value is larger.

Figure 6 shows the clustering results of parameter α and feature# in Cora dataset. As depicted in Figure 6a, when parameters α and feature# increase, the ACC value increases. In Figure 6b, feature# is sensitive to NMI value, while the overall relative value of feature# is larger and the NMI value is larger.

Figure 7 depicts the clustering results of parameters β and feature# in the Cora dataset. As depicted in Figure 7a, for β≤0.01, the ACC increases as feature# increases; otherwise, for β>0.01 the ACC value remains basically unchanged. Figure 7b shows that, when feature#=100 and 500, the NMI value is larger.

Figure 8, Figure 9 and Figure 10 show the clustering results of parameters α, β, and feature# in CiteSeer dataset. Figure 8a shows that the ACC is insensitive to parameters α and β, but Figure 8b shows that the NMI is insensitive to parameters α∈{0.001,0.01,1,10,100} and β∈{0.01,0.1,1,100}; when the two parameters are larger or smaller, the NMI value exhibits a small fluctuation.

Figure 9 shows the clustering results of parameters α and feature# in the CiteSeer dataset. As illustrated in Figure 9a, the magnitude of the ACC value difference is 0.05 and the ACC is insensitive to parameter α. For feature#=300, the ACC has better performance. As shown in Figure 9b, the NMI is slightly sensitive to parameters α and β. For α=1, when feature#=200, a larger NMI is achieved.

Figure 10 shows the clustering results of parameters β and feature# in the CiteSeer dataset. As demonstrated in Figure 10a, the magnitude of the ACC value difference is 0.05 and the ACC is insensitive to parameter β. In general, the NMI performance is better when β∈{0.001,0.01,1}. The ACC performance is stabler when feature#>200. Figure 10b shows that the NMI results are almost insensitive to parameter β and feature# in the CiteSeer dataset. When β=0.001, the NMI result is greater.

Figure 11, Figure 12 and Figure 13 show the clustering experiment results of parameters α, β, and feature# in the BBCSport dataset.

Figure 11a shows that the ACC is insensitive to parameters α and β in the BBCSport dataset. Figure 11b shows that the NMI is insensitive to parameter α but the NMI changes slightly when β≥1. However, the NMI results have a peak when β=1.

Figure 12a shows the clustering ACC of parameters α and feature# in the BBCSport dataset. The magnitude of the ACC value difference in this figure is 0.05, and the ACC is insensitive to parameter α and feature# in BBCSport dataset. Comparatively, it has high ACC with feature#≥400 and 0.1≤a≤10. Figure 12b shows the clustering NMI of parameters α and feature# in the BBCSport dataset. When feature#≤200, the results of NMI are insensitive to parameters α and feature#. NMI increases first and then decreases with parameter α. When feature#>200, NMI has a greater value when a=0.1 and feature#=300 and 400.

Figure 13a shows the clustering ACC of parameters *b* and feature# in the BBCSport dataset. The magnitude of ACC value difference in this figure is 0.05, and the ACC is insensitive to parameters β and feature# in the BBCSport dataset. Comparatively, it has high ACC with feature#≥300 and *a* = 10,000. Figure 13b shows the clustering NMI of parameters *b* and feature# in the BBCSport dataset. NMI is sensitive to feature#, and NMI has a greater value when feature#≥300 and b≥100.

Figure 14, Figure 15 and Figure 16 show the clustering results of parameters α, β, and feature# in BlogCatalog dataset.

Figure 14a shows that ACC is insensitive to α but, when β=1000, its performance is better. Figure 14b shows the NMI performance for parameters α and β. NMI is not very sensitive to α and β, and, when β is larger and α is smaller, NMI is relatively larger.

Figure 15a shows the ACC performance with parameters α and feature#. When feature#=1500 and α>10, the ACC performance is better. Figure 15b shows the NMI decrease with parameter feature# and NMI is not sensitive to α. Figure 15 indicates that higher feature# is not necessarily better for BlogCatalog data.

Figure 16 shows that β is sensitive to ACC and NMI, and feature# is sensitive to clustering performance. In Figure 16a, when feature#≥1500 and β≥1, the ACC is better. In Figure 16b, NMI increases with parameter β; when β≥10, NMI has a greater value.

Parameter sensitivity remains a challenging and unsolved problem in feature selection. This experiment analyzes the sensitivity of parameters α, β, and feature#. We performed similar parameter sensitivity analyses for the data sources. The results show that MFSC is almost insensitive to parameters α and β for ACC performance. This shows the importance of capturing the multiview manifold structure embedded in the feature selection model. However, the MFSC is sensitive to feature#. This is because the network size affects the number of feature selections.

### 4.7. Convergence Analysis

The convergence effects of the datasets are shown in Figure 17. In addition, the convergence effect of the remaining data is similar. Based on the experimental results, the convergence effect is relatively good. The objective function increases as the number of convergences increases and quickly reaches a constant convergence value regardless of the initial objective value.

## 5. Conclusions and Future Work

This study proposes a multiview clustering-guided feature selection algorithm for multiview data, which integrates subspace learning and feature selection, and embeds the norm of manifold regularization. This feature selection algorithm reduces the influence of redundancy and the irrelevant matrix of the multiview data. In addition, clustering is used as the standard for feature selection. This algorithm can perform feature selection to ensure that the clustering structure remains unchanged. It is noteworthy that the complementary contribution of each view is fully considered. The optimization process is calculated and theoretically analyzed, and experiments are performed using a multiview dataset. It can be concluded that the algorithm is effective and superior to many existing feature selection algorithms or multiview clustering algorithms.

Although our method achieves good clustering performance, on the one hand, we mainly consider social network data, while other types of multimodal data graph structures are not considered. On the other hand, some parameters need to be manually adjusted. Recently, deep learning has demonstrated excellent feature extraction capabilities in multiview data, such as images and natural languages. In the future, we will study how to integrate deep learning and the MFSC model to process multiview data and accurately describe semantic information. 

## Figures and Tables

**Figure 1 entropy-25-01606-f001:**
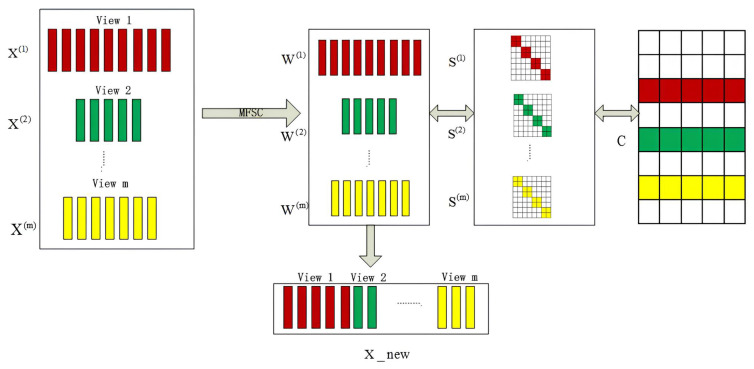
Overall framework based on the parallel mode in MFSC.

**Figure 2 entropy-25-01606-f002:**
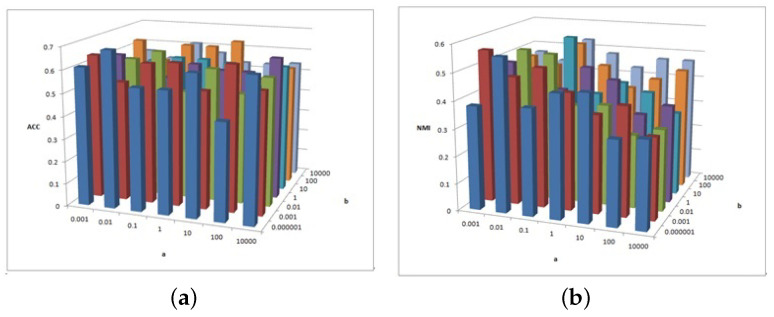
(**a**) ACC values of parameter α and parameter β for 3sources data. (**b**) NMI values of parameter α and parameter β for 3sources data.

**Figure 3 entropy-25-01606-f003:**
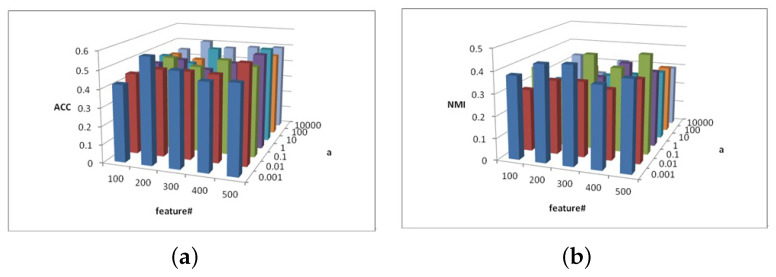
(**a**) ACC values of parameter α and parameter feature# for 3sources data. (**b**) NMI values of parameter α and parameter feature# for 3sources data.

**Figure 4 entropy-25-01606-f004:**
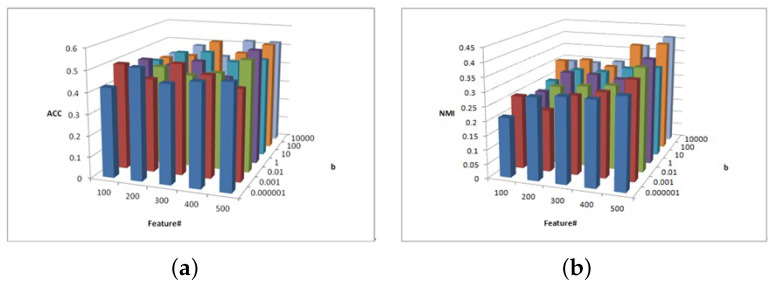
(**a**) ACC values of parameter β and parameter feature# for 3sources data. (**b**) NMI values of parameter β and parameter feature# for 3sources data.

**Figure 5 entropy-25-01606-f005:**
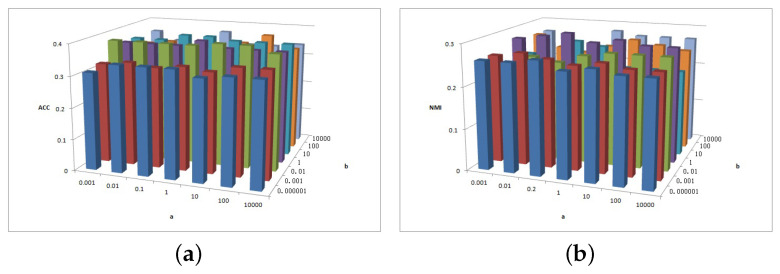
(**a**) ACC values of parameter α and parameter β for Cora data. (**b**) NMI values of parameter β and parameter feature# for Cora data.

**Figure 6 entropy-25-01606-f006:**
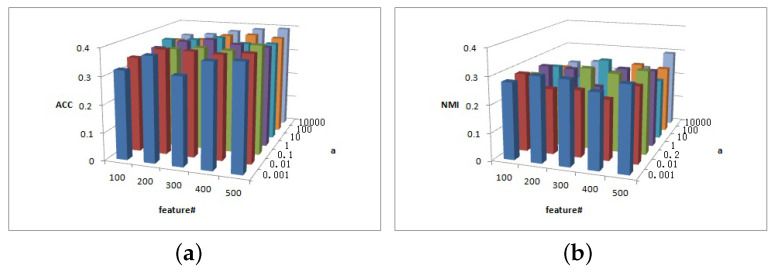
(**a**) ACC values of parameter α and parameter feature# for Cora data. (**b**) NMI values of parameter β and parameter feature# for Cora data.

**Figure 7 entropy-25-01606-f007:**
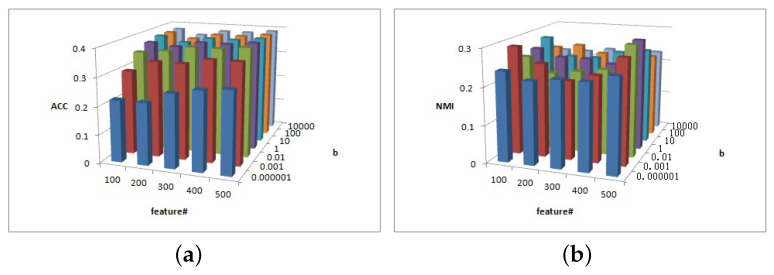
(**a**) ACC values of parameter β and parameter feature# for Cora data. (**b**) NMI values of parameter β and parameter feature# for Cora data.

**Figure 8 entropy-25-01606-f008:**
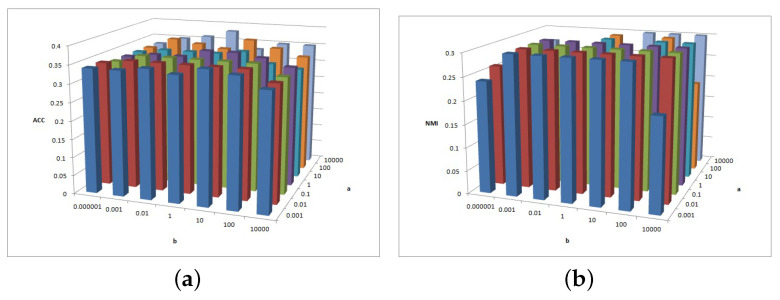
(**a**) ACC values of parameter α and parameter β for CiteSeer data. (**b**) NMI values of parameter α and parameter β for CiteSeer data.

**Figure 9 entropy-25-01606-f009:**
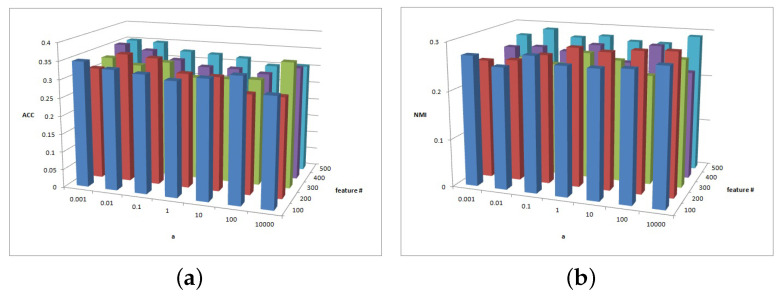
(**a**) ACC values of parameter α and parameter feature# for CiteSeer data. (**b**) NMI values of parameter α and parameter feature# for CiteSeer data.

**Figure 10 entropy-25-01606-f010:**
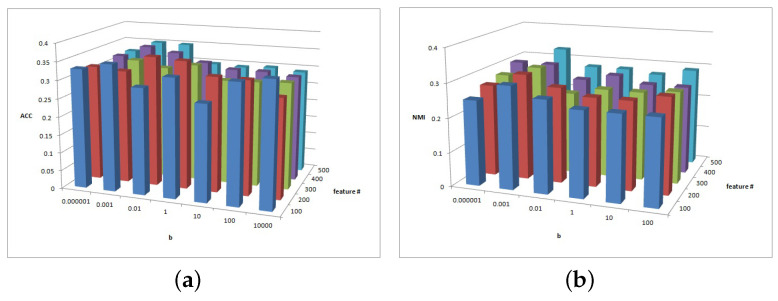
(**a**) ACC values of parameter β and parameter feature# for CiteSeer data. (**b**) NMI values of parameter β and parameter feature# for CiteSeer data.

**Figure 11 entropy-25-01606-f011:**
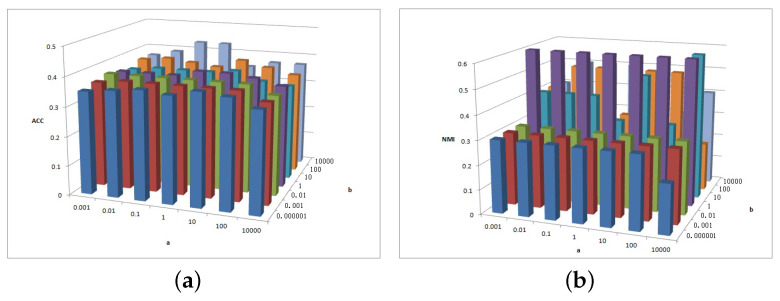
(**a**) ACC values of parameter α and parameter β for BBCSport data. (**b**) NMI values of parameter α and parameter β for BBCSport data.

**Figure 12 entropy-25-01606-f012:**
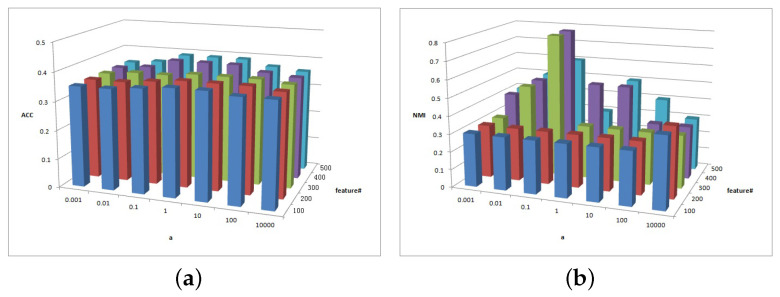
(**a**) ACC values of parameter α and parameter feature# for BBCSport data. (**b**) NMI values of parameter α and parameter feature# for BBCSport data.

**Figure 13 entropy-25-01606-f013:**
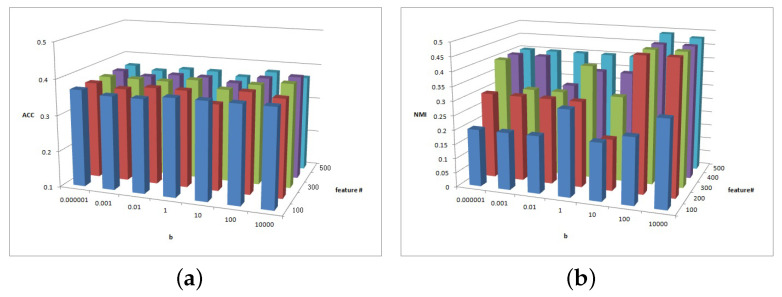
(**a**) ACC values of parameter β and parameter feature# for BBCSport data. (**b**) NMI values of parameter β and parameter feature# for BBCSport data.

**Figure 14 entropy-25-01606-f014:**
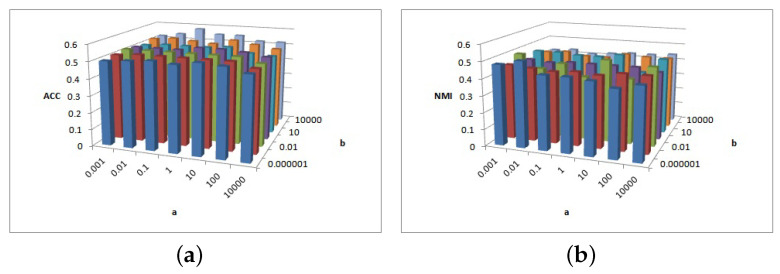
(**a**) ACC values of parameter α and parameter β for BlogCatalog data. (**b**) NMI values of parameter α and parameter β for BlogCatalog data.

**Figure 15 entropy-25-01606-f015:**
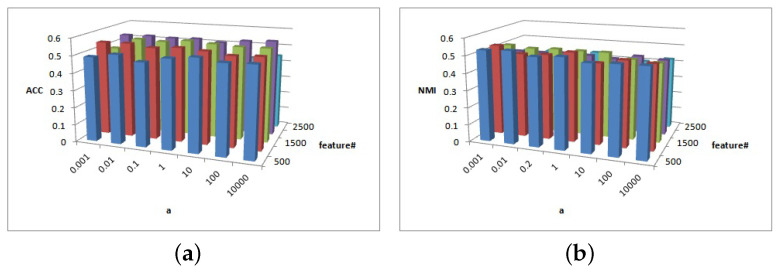
(**a**) ACC values of parameter α and parameter feature# for BlogCatalog data. (**b**) NMI values of parameter α and parameter feature# for BlogCatalog data.

**Figure 16 entropy-25-01606-f016:**
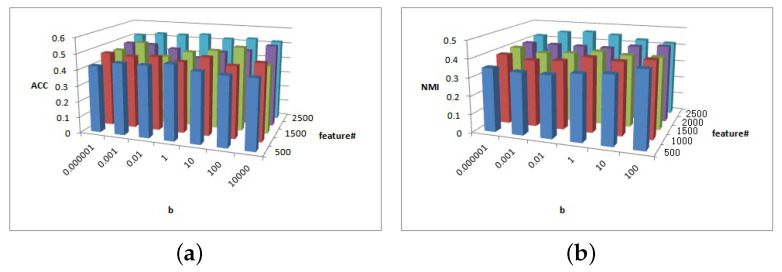
(**a**) ACC values of parameter β and parameter feature# for BlogCatalog data. (**b**) NMI values of parameter β and parameter feature# for BlogCatalog data.

**Figure 17 entropy-25-01606-f017:**
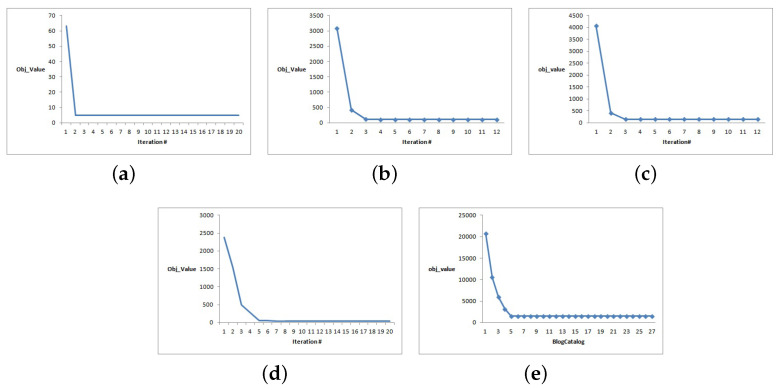
MFSC convergence curves. (**a**) 3sources; (**b**) Cora; (**c**) CiteSeer; (**d**) BBCSport; (**e**) BlogCatalog.

**Table 1 entropy-25-01606-t001:** Statistical table of typical datasets.

	3sources			Cora		CiteSeer		BBCSport		BlogCatalog	
the number of vertexes	169			2708		3312		544		10,312	
the number of classes	6			7		6		5		6	
the number of views	3			2		2		2		2	
view	V1	V2	V3	V1	V2	V1	V2	V1	V2	V1	V2
the number of features	3560	3631	3068	1433	2708	3703	3312	3183	3203	6115	5764

**Table 2 entropy-25-01606-t002:** ACC of different methods on typical datasets.

	3sources	Cora	CiteSeer	BBCSport	BlogCatalog
LapScore	0.36	0.3	0.32	0.39	0.55
RelieF	0.42	0.27	0.31	0.42	0.59
MCFS	0.55	0.29	0.27	0.47	0.54
PRMA	0.54	0.26	0.36	0.37	0.56
GMNMF	0.46	0.25	0.26	0.45	0.57
SCFS	0.65	0.4	0.34	0.45	0.56
JMVFG	0.64	0.35	0.44	0.42	0.52
CCSFS	0.54	0.3	0.4	0.45	0.5
MFSC	0.69	0.39	0.38	0.48	0.54

**Table 3 entropy-25-01606-t003:** NMI of different methods on typical datasets.

	3sources	Cora	CiteSeer	BBCSport	BlogCatalog
LapScore	0.18	0.27	0.3	0.15	0.32
RelieF	0.35	0.25	0.5	0.32	0.4
MCFS	0.48	0.27	0.28	0.24	0.29
PRMA	0.49	0.24	0.34	0.42	0.31
GMNMF	0.4	0.29	0.56	0.51	0.33
SCFS	0.54	0.35	0.4	0.6	0.47
JMVFG	0.45	0.3	0.51	0.57	0.42
CCSFS	0.31	0.34	0.41	0.56	0.4
MFSC	0.58	0.3	0.42	0.78	0.45

## Data Availability

Data are contained within the article.

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
