# Peer review of "Multiview Data Clustering with Similarity Graph Learning Guided Unsupervised Feature Selection"

_entropy, 2023, doi:10.3390/e25121606_

Round 1
Reviewer 1 Report
Comments and Suggestions for Authors
The used datasets have a small size. Some large ones should be tested to show its effectiveness. For example, Large-scale Multi-view Subspace Clustering in Linear Time.
The compared methods are quite old. Some recent methods should be adopted. For example, Structured Graph Learning for Scalable Subspace Clustering: From Single-view to Multi-view, High-order Multi-view Clustering for Generic Data.
The figures can be improced.
Comments on the Quality of English LanguageNone
Author Response
Dear Editor and Review:
Thank you for your letter and for the reviewers’ comments concerning our manuscript entitled “Multiview Data Clustering with Similarity Graph Learning Guided Unsupervised Feature Selection” (ID: entropy-2708307). Those comments are all valuable and very helpful for revising and improving our paper, as well as the important guiding significance to our researches. We have studied comments carefully and have made correction which we hope meet with approval. The main corrections in the paper and the responds to reviewer’s comments as flowing:
1.Response to comment: The used datasets have a small size. Some large one should be tested to show its effectiveness.
Response: We have added experiments on large-scale data :BlogCatalog. The data consists of 10312 articles, divided into 6 categories, each article with two views: blog content and its related tags. The experimental results are shown in Figure 14- 16 and Table 1-2.
2.Response to comment: The compared methods are quite old. Some recent methods should adopted.
Response: We have added two compared methods: JMVFG (2022) and CCSFS (2023), The experimental results are shown in Table1-2. Two methods compared the clustering ACC and NMI of 3sources, Cora, Citeseer, BBCSport and BlogCatalog dataset.
3.Response to comment: Dose the introduction provide sufficient background and include all relevant reference? Can be improved?
Response: We have supplemented the third paragraph of the Introduction section adding the current researches of unsupervised feature selection. The content is as follows “Zhang et al.[10] propose a formulation that ……for the purpose of feature selection.“ in page 2, and we added 5 recent feature selection references.
- Response to comment: Are the conclusion supported by the results? Can be improved?
Response: Joining large-scale data BlogCatalog and recent methods JMVFG (2022) and CCSFS (2023) for experiments, the experimental results (Table 1-2, Figure 14-17) show that our method has some advantages in cluster learning and feature selection.
5.Response to comment: Minor editing of English required.
Response: Solve English capitalization issues, such as changing “Where” to “where”, punctuation and space issues, such as “PRMA[7](Pro” to “PRMA[7] (Pro..”, symbol representation errors, such as “the j-th diagonal element ..” to “the $j$-th diagonal element..” etc

Reviewer 2 Report
Comments and Suggestions for Authors
The paper presents a multiview data clustering algorithm, MFSC (Multiview Feature Selection Clustering), which integrates similarity graph learning with unsupervised feature selection. The paper is well-written and the proposed work sounds technically and scientific and addresses challenges like high dimensionality and redundancy in multiview data. I just have a few comments.
-Authors presented a detailed systematic evaluation of the Multiview Feature Selection Clustering (MFSC) algorithm through various experiments and comparisons with other related algorithms which proves that the proposed work outperforms similar other works in the literature.
-The comparison results with algorithms like LapScore, Relief, MCFS, PRMA, GMNMF, and SCFS are provided. However, are these algorithms the most relevant or current in the field? Authors need to justify their choices for comparative evaluations.
-The results are good, but the paper can further discuss how generalizable these results are across different types of multiview data not covered in the study.
-Consider using vector graphics (e.g., EPS, PDF) for images to maintain high quality for publication. I also suggest using 2D plots instead of 3D plots due to the type of data, which can help make the results graphs more readable.
N/A
Reviewer 3 Report
Comments and Suggestions for Authors
The manuscript presents a novel clustering algorithm for multiview data, termed Multiview Feature Selection Clustering (MFSC). This algorithm aims to leverage the consistent or complementary information present in multiview data to enhance clustering results. The high dimensionality, lack of labels, and redundancy in multiview data pose significant challenges to clustering, which the MFSC algorithm seeks to address. By integrating local manifold regularization into similarity graph learning and using clustering labels as a standard for unsupervised feature selection, MFSC maintains the manifold structure of multiview data while retaining the characteristics of the clustering label. The algorithm's effectiveness is demonstrated through systematic evaluations using benchmark multiview and simulated data, showing that MFSC outperforms traditional algorithms in clustering experiments.
Strengths:
• Innovative Approach: The manuscript introduces an innovative clustering algorithm that combines similarity graph learning with unsupervised feature selection, a unique approach that addresses the challenges of multiview data clustering.
• Effective Integration: Integrating local manifold regularization into similarity graph learning is a novel aspect that helps preserve the various structures of multiview data.
• Empirical Validation: The algorithm is systematically evaluated using benchmark multiview and simulated data, providing empirical evidence of its effectiveness compared to traditional algorithms.
Concerns:
How does the computational complexity of the MFSC algorithm compare to traditional clustering algorithms?
Can the authors discuss any limitations of the MFSC algorithm and potential areas for future improvement?
Literature Review: Some related works are missing, for instance, Graph self-representation method for unsupervised feature selection, Unsupervised feature selection for visual classification via feature-representation property, Unsupervised spectral feature selection with dynamic hyper-graph learning, Sparse sample self-representation for subspace clustering, and Low-rank feature selection for multi-view regression
Minors:
After the equation, "Where" should be "where";
"4) PRMA [7](Probabilistic robust matrix approximation )" -> "4) PRMA [7] (Probabilistic robust matrix approximation)";
"…a diagonal matrix and the j-th diagonal element is …" -> "…a diagonal matrix and the $j$-th diagonal element is …" ;
I just picked out a few minor errors, there are many more in the article, and a carefully proofread version is needed.
Overall, the paper is good and interesting.
Author Response
Dear Editor and Review:
Thank you for your letter and for the reviewers’ comments concerning our manuscript entitled “Multiview Data Clustering with Similarity Graph Learning Guided Unsupervised Feature Selection” (ID: entropy-2708307). Those comments are all valuable and very helpful for revising and improving our paper, as well as the important guiding significance to our researches. We have studied comments carefully and have made correction which we hope meet with approval. The main corrections in the paper and the responds to reviewer’s comments as flowing:
1.Response to comment: After the equation, "Where" should be "where";"4) PRMA [7](Probabilistic robust matrix approximation )" -> "4) PRMA [7] (Probabilistic robust matrix approximation)"; "…a diagonal matrix and the j-th diagonal element is …" -> "…a diagonal matrix and the $j$-th diagonal element is …" .
Response:Solve English capitalization issues, such as changing “Where” to “where”, punctuation and space issues, such as “PRMA[7](Pro” to “PRMA[7] (Pro..”, symbol representation errors, such as “the j-th diagonal element ..” to “the $j$-th diagonal element..”, Singular and plural errors, such as “shows…” to “show”, “coef?cient” to “coefficient”, “?xed” to “fixed” etc..
2. Response to comment: How does the computational complexity of the MFSC algorithm compare to traditional clustering algorithms?
Response: Analyzed the time complexity of several clustering algorithms, as follows:
JMVFG(reference 11): time complexity $O\sum^v_{i=1}[(m^{(i)}^3+(m^{(i)}^3n+m^{(i)}n^2))$
MCFS(reference:24): time complexity $O(n^2m)$
Kmeans: time complexity $O(IKmn)$
Spectral clustering: time complexity $O(n^3))$
MFSC: time complexity $O(\sum _{v=1}^m (d^{(v)}\times n\times k+n^2 \times k) )$.
Our time complexity is consistent with recent multimodal clustering algorithms and superior to spectral clustering and so on.
- Response to comment: Can the authors discuss any limitations of the MFSC algorithm and potential areas for future improvement?
Response: The “Conclusion” section has added limitations and future developments of MFSC. The content is as follows “Although our method achieves good clustering performance,….. semantic information.” in page 18
4.Response to comment: Literature Review: Some related works are missing, Graph self-representation method for unsupervised feature selection, Unsupervised feature selection for visual classification via feature-representation property, Unsupervised spectral feature selection with dynamic hyper-graph learning, Sparse sample self-representation for subspace clustering, and Low-rank feature selection for multi-view regression
Response: We have supplemented the third paragraph of the Introduction section adding the current researches of unsupervised feature selection. The content is as follows “Zhang et al.[10] propose a formulation that learns an adaptive……Liu et al.[14] propose a framework for guided unsupervised feature selection, “ in page 2
